# *Migrant Women’s Health and Safety:* Why Do Ethiopian Women Choose Irregular Migration to the Middle East for Domestic Work?

**DOI:** 10.3390/ijerph192013085

**Published:** 2022-10-12

**Authors:** Zewdneh Shewamene, Cathy Zimmerman, Eyasu Hailu, Lemi Negeri, Annabel Erulkar, Elizabeth Anderson, Yuki Lo, Orla Jackson, Joanna Busza

**Affiliations:** 1Department of Public Health, Environment & Society, Faculty of Public Health and Policy, London School of Hygiene and Tropical Medicine, Tavistock Place, London WC1H 9SH, UK; 2Department of Global Health and Development, Faculty of Public Health and Policy, London School of Hygiene and Tropical Medicine, 15-17 Tavistock Place, London WC1H 9SH, UK; 3Population Council, Heritage Plaza, Bole Medhaneialem Road, Addis Ababa 18609, Ethiopia; 4The Freedom Fund, Lighterman House, 30 Wharfdale Road, London N1 9RY, UK

**Keywords:** irregular migration, domestic labour, women, Ethiopia, Middle East

## Abstract

Background: Low-wage labour migration is an increasing determinant of global health, associated with risks of exploitation, abuse, and unsafe conditions. Despite efforts to prevent irregular migration and initiatives to warn individuals of the risks of trafficking, many migrants still opt for irregular channels, particularly women seeking jobs as domestic workers. Ethiopia is one of the largest source countries for female migrants entering the domestic labour market in the Middle East. This qualitative study explored migration decision making by Ethiopian women traveling to the Middle East for domestic labour, focusing on the use of irregular channels. Methods: We conducted semistructured interviews with policy stakeholders, migration recruiters, and returnee domestic workers. Results: We identified three main themes that help explain decision making by female migrants and their communities. First, women were not always clear whether they were using legally approved processes, particularly because of the range of individuals involved in arranging migration plans. Second, irregular migration was seen to be quicker and easier than regular migration procedures. Third, study participants believed the risks between irregular and regular migration were similar. Conclusion: Our study highlights challenges associated with antitrafficking initiatives that discourage irregular migration and suggests new perspectives to address the health risks linked to labour migration.

## 1. Introduction

Irregular labour migration is a global phenomenon with growing implications for the health of both migrating and host populations [1]. The International Labour Organization (ILO) estimated that there were 169 million labour migrants in 2019, comprising five percent of the global labour force and thus representing an integral part of the world economy [2]. Many migrant workers are engaged in informal sectors that expose them to unsafe living and working conditions [2,3]. Participation in low-wage labour is associated with substantial health and safety risks, particularly abuses associated with human trafficking and labour exploitation [4,5]. The United Nations 2030 Sustainable Development goals highlight the importance of ensuring safe migration and protecting the labour rights of all workers, with particular attention to women migrants [6]. 

Globally, figures suggest that there are approximately 11.5 million migrant domestic workers, most of whom are women [7]. The Middle East is a common destination for female migrants from low-income countries in Africa and Asia seeking domestic work [8,9,10,11,12,13,14]. Labour migrants to the Middle East tend to be low-skilled, poorly educated, and often unaware about work conditions, which increase their risk of experiencing various forms of exploitation, including underpayment, excessive working hours, and physical or verbal abuse [12,14]. Unsafe work conditions can lead to injuries and long-term morbidity, especially compared to non-migrants working in the same sectors [15]. There is a growing body of literature highlighting domestic workers’ exposure to health risks, including occupational hazards, physical and sexual abuse, and other human rights violations, all of which contribute to poor mental and physical health outcomes [8,16,17,18]. 

Ethiopia has become one of the largest providers of female migrants entering the domestic labour market in the Middle East [8,9,19]. Health risks experienced by Ethiopian migrant domestic workers have been well-documented, and include occupational injuries, unwanted pregnancies, emergencies (due to accidents, suicide attempts, or assaults), sexual violence, and chronic illnesses such as cancer and tuberculosis exacerbated by inadequate access to treatment [20,21]. To try to protect its citizens, the Ethiopian government imposed a ban on labour out-migration in 2013, which it subsequently revoked in 2018 and replaced with “safer migration” policies known as Overseas Employment Proclamations, centred around the regulation of overseas employment agencies [22]. Nonetheless, many Ethiopians have continued to depart irregularly, both during and since the ban [13].

The Ethiopian policy reflects global efforts to prevent irregular migration. For both beneficent motives, i.e., to protect individuals from trafficking, and for political and national security reasons, migration initiatives aim to deter individuals from using unofficial routes to migrate for work, emphasising that these are inherently less safe [23]. Yet, emerging findings suggest that there may not be significant differences in risk between individuals who migrate via official channels and those who migrate using informal routes, for example, relying on informal agents or family members [23]. To what extent these policies align with migrants’ own priorities and concerns has not been explored, and there has similarly been very limited research on how individuals decide whether to use formal versus informal processes or how this decision might impact their risk of harm. As countries continue to promote regular migration, it is essential to understand what women migrants themselves consider best for their future employment, health, and safety. 

This study contributes to filling these gaps by examining migrant women’s decisions about *how* to migrate, using the case of Ethiopian female migrants to the Middle East. In particular, the study examines women’s use of irregular or informal migration channels, defined as the movement of Ethiopian female domestic workers to Middle East countries without fulfilling the requirements set by the Overseas Employment Proclamations. The findings are intended to provide evidence to help inform realistic programming on safe out-migration for domestic work.

## 2. Methods 

### 2.1. Study Design and Setting 

The study used semistructured qualitative interviews to elicit individual opinions, experiences, and perspectives of respondents, including: (1) national and subnational stakeholders; (2) formal migration recruiters (registered agencies); and (3) Ethiopian domestic workers who had recently returned from Lebanon. Trained qualitative researchers followed a topic guide specifically designed for each type of respondent. Interviews were carried out over the phone or in the offices of the stakeholders and recruiters, and at a rehabilitation centre where returnee domestic workers stayed following their return to Ethiopia. Interviews lasted approximately 60–90 min and were undertaken in Amhara or Afaan Oromo, the two most commonly spoken languages in Ethiopia. Topic guides addressed respondents’ involvement in migration-related policy or programming, knowledge or personal experience of using different kinds of migration recruiters, and views on facilitators and barriers to migrants’ safety.

### 2.2. Sampling and Recruitment 

Purposive and snowball sampling were used to identify participants. Recruitment procedures varied by type of respondent. First, a list of migration-related policy stakeholders was drawn up by the research team, drawing on existing knowledge and professional networks. Interviews were conducted with 18 stakeholders, representing a mix of institutions working on migration and labour issues, including international organisations (UN agencies and development and humanitarian organisations), government ministries and agencies at federal and regional level, and national civil society associations. 

Second, the sampling strategy for recruiters was designed to select private overseas employment agencies listed on a noticeboard at the Ministry of Labour and Social Affairs (MOLSA) office (thus, by definition, formally licensed) followed by peer referrals. Although over 800 overseas employment agencies were registered with MOLSA, we were able to identify only 400 from the MOLSA noticeboard and other sources. Of these, 23 were contacted by phone or through in-person visits to their work premises and nine agreed to be interviewed. An additional 12 respondents were identified through peer referral and all agreed to an interview. Topic guides for recruiters focused on how they identify potential migrants, their role in facilitating activities, and their views on determinants of risk for women seeking domestic work in Middle East countries. 

Finally, 27 returnees from Lebanon were interviewed following their repatriation to Ethiopia with assistance from AGAR (a local charity with a rehabilitation centre) and the Freedom Fund, an international antislavery organization supporting migrants’ rights in Ethiopia. This particular group of returnees required assistance to return to Ethiopia following economic collapse and COVID-related travel restrictions in Lebanon, which left them homeless and unable to arrange their own travel. After arriving in Addis Ababa, returnees received support over a few days at AGAR’s rehabilitation shelter until being reunited with their families. Individuals with signs of emotional distress were assessed by a psychologist and invited to stay for several weeks in the shelter to receive counselling support. For this research, only those women assessed as psychologically healthy were recruited for interview. Three trained female fieldworkers conducted interviews at the shelter. Interviewers received specialised trauma-informed training focused to ensure the women participated voluntarily and interviewers were instructed on safeguarding measures for participants who might become emotionally distressed or agitated when recalling upsetting experiences in Lebanon or during the repatriation process. A psychologist was available on location and voluntary referrals were made for counselling for respondents who requested assistance or demonstrated distress. 

### 2.3. Data Analysis 

Interview data were collected using a digital audio recorder and notes were taken by fieldworkers during and immediately following the interview. All audio-recorded interviews were transcribed in full by fieldworkers and professional transcribers. Interviews were conducted in Amharic or Afaan Oromo and translated into English. Then, two researchers (one from the UK and one Ethiopian) undertook data familiarisation by repeatedly rereading the transcripts. Data checking was conducted by the Ethiopian researcher who listened to the recoded audio interviews while reviewing transcripts to ensure accuracy of translation and provide quality control for transcriptions. Through a process of data immersion (carefully reading and rereading each interview transcript), researchers reflected on the entirety of the data set to understand and interpret the views of respondents about responsible migration in general. This review was followed data-driven thematic analyses using NVivo software to identify themes and conceptual categories, and highlight commonalities and differences in participants’ responses.

### 2.4. Ethical Consideration 

All respondents provided either verbal or written informed consent. Requests for consent emphasised that information provided would remain anonymous, that respondents could terminate the interview and withdraw their consent at any time, and that any data shared in reports or other research outputs would be anonymised to prevent identification of respondents or individuals they may have named. Returnees were further assured that refusal to participate in the study would in no way affect their access to services received at the rehabilitation centre. All respondents were over the age of 18 and were not paid or given incentives for their participation in the study.

The names of individuals and/or institutions have not been reported to protect the identity of participants. The included excerpts from interviews are identified using the category of the respondent and numbers to avoid accidental breach of anonymity (e.g., returnee #1, recruiter #2, stakeholder #3, etc.). 

## 3. Results

In total, 66 individual interviews were conducted comprising 18 policy stakeholders, 21 recruiters, and 27 returnee migrants from Lebanon. Policy stakeholders were diverse in terms of their roles vis à vis ensuring safer migration of low-skilled domestic worker migrants from Ethiopia to the Middle East corridor. Recruiters were representatives of registered foreign employment agencies and returnee participants were all female low-skilled domestic workers who were repatriated from Lebanon (Table 1).

In this paper, we present findings on the reasons why Ethiopian women who migrate for domestic work travel through “irregular” means despite warnings of potential risks. We compared perspectives across the different categories of respondents and identified three themes that capture the migration decision-making process as follows: (1) ease and speed, (2) perceptions of risk, and (3) difficulty distinguishing regular from irregular migration.

### 3.1. Ease and Speed 

Participants explained that prospective migrants may knowingly choose irregular migration to save time, as regular migration can be time-consuming and bureaucratic. The regular migration process usually takes more than three months and requires women to attend trainings for 1–2 months. If women wish to avoid these procedures, they select informal options that, for example, use tourist visas to enter the destination country that are quicker to obtain. As recruitment agents explained:

*The illegal process done by [irregular] brokers is faster than the process done by licensed agencies. Because the licensed agencies take more time to finalise the legal process, it may take 1 to 2 months, but the brokers finish the process within 10 to 15 days only*.(Recruiter #17)


*The migrants prefer illegal agencies to licensed overseas employment agencies because the former do things very quickly. But with us [registered recruiters] it is a long process and complicated*
(Recruiter #2)

Recruiters operating outside the official system can also avoid dealing with the Ministry of Labour and Social Affairs (MOLSA) and Ethiopian embassies in destination countries, which are considered bureaucratic, as described by another recruiter.


*The bureaucratic hurdles within MOLSA will also prolong the process. It takes around three months for the Ethiopian Embassy to Saudi Arabia to examine the migrant’s documents for their authenticity and approve the employment contract*
(Recruiter #8)

Furthermore, migrants considered some of the Overseas Employment Proclamation eligibility criteria difficult to meet. Migrants were legally required to complete predeparture training and, at the time of data collection, to prove attainment of at least eighth grade education. The latter stipulation meant that women who had migrated in the past under a different legal framework, then returned to Ethiopia, found they were now ineligible to remigrate, as the proclamation had been introduced during or after they had worked abroad. Several stakeholders and recruiters pointed out that returnees were most experienced and prepared for domestic work but had become the least able to meet the prerequisites. Thus, returnee domestic workers who knew how to navigate the system in different countries and had learned basic Arabic, but had not attended school through eighth grade, found that their only option was to travel through irregular means, using unregistered facilitators. In this way, the first Overseas Employment Proclamation (923/2016) was seen to drive remigrating women into the hands of individuals who might exploit them: 


*The law says that the migrant should be a graduate of 8th grade to be eligible to migrate to Arab countries in search of a job. Let me tell you my own experience in this regard. There was one young Ethiopian woman lived and worked in Arab countries for ten years. Then, she returned to her country because of some problems. Now she is planning to return [abroad]. However, she did not complete 8th grade education. Therefore, she will use [unregistered] brokers*
(Recruiter #10)

Respondents suggested that this and other requirements should be waived for migrants who had prior experience as domestic workers in the Middle East. In 2021, a new Overseas Employment Proclamation was issued that removed the educational requirement. However, first-time migrants must still undergo predeparture training, the content of which was criticized by a few respondents. One highlighted that women who were previously employed in domestic work did not gain much from the training, while another felt some of the content was generally inappropriate: 


*They require predeparture trainings for all [migrants] including those girls who have a lot of experience in an Arab country. They already know how things are done there from their own experience. In my opinion, there’s no need to make those who already know take the training, it is better to educate the ones who come from the rural area of Ethiopia and who are new for traveling*
(Recruiter #21)

*To our dismay and to the dismay of the migrants, the trainees are shown how to serve wine and other types of alcoholic drinks. But the reality on the ground is that consumption of alcohol is strictly prohibited in most Arab countries*.(Recruiter #7)

Because migrants do not value these trainings and because they are in a rush to finalise their migration once they decide to leave, they are frustrated by any additional time or bureaucracy. These perceived hurdles make irregular migration more attractive.

### 3.2. Perception That Migration Risks Are Not Decreased through Regular Migration

Many participants indicated that although the terms “legal” and “illegal” are used to signify “safe” versus “unsafe” migration, in reality, migration outcomes are poorly predicted by the choice of regular or irregular means. In the current reality of women’s migration from Ethiopia to Middle East countries, few recruiters are likely to protect migrant workers. Respondents suggested that agencies offer little more than help with visa paperwork and transportation, regardless of whether they are licensed or unlicensed. As one returnee described, there are risks related to legal versus illegal brokers: 


*I think one must be fortunate. Whether one uses the legal or illegal means to go to Arab countries, ultimately our safety is determined by the behaviour of our employers. We will be at a risk of being harmed if our employers have violent and threatening behaviour. Actually, there is no difference between legal and illegal migration*
(Returnee #10)

Policy stakeholders reflected similar opinions when discussing the limited protection provided by all types of recruiters:


*There are no recruiters who are protecting migrants, almost all of them are risky. Most of the time recruiters are not worrying about the safety and wellbeing of migrants. Since recruiters are committing modern-time slavery and they mostly worrying about getting money, they are considering migrants as if they are transporting goods*
(Stakeholder #1)

Respondents felt that both licensed and unlicensed recruiters failed to protect migrants because they compete with one another for profits. Stakeholders explained that recruiters show general disregard for migrants and even pressure returnees not to damage their reputations with negative stories about their experiences: 


*Recruiters … even come to rehabilitation centres to convince returnees not to testify against them, let alone participating in [migrants’] protection. They are busy planning to send them again to the Middle East. There are a few legal recruiters who may protect migrants, but I can’t say all legal facilitators participate [in protection].*
(Stakeholder #17)

Moreover, even when recruiters promised to support migrants if they experienced problems abroad, they did not seem to adhere to this promise, as recalled by one returnee: 

*Generally, facilitators will tell you to call [them] if you have any problem. But in reality, they are not giving a reply to anyone. I called the broker in Ethiopia and asked him to call the broker in the destination to find me a job somewhere else. He advised me to flee the house. He told me that I must pretend to be throwing something away [i.e., rubbish] when I flee the house. But it was impossible for me to escape in that way because of the security camera hung on the wall. I called him for the second time and told him about my decision to return to my country by my own expense as soon as possible. But his mobile phone did not work then after*.(Returnee #6)

Corrupt practices by recruiters were also mentioned, such as registered agencies using their license as a cover to send migrants through irregular channels similar to unregistered recruiters. 

*Not only illegal recruiters but legal recruiters also follow the shortest path rather than the right procedure … for their own financial gain*.(Stakeholder #12)


*There are different agencies, some of them [are] working legally as well as informally by establishing a long chain starting from the broker who recruited migrants from the rural area up to abroad. … Since there are owners of the agencies working in the sector who are working illegally, I couldn’t see a proper way of doing things in this sector.*
(Recruiter #17)

However, regular migration was seen as cheaper than migrating through irregular means. Most of the travel costs associated with regular migration are borne by the employers, while those who migrate irregularly cover all their expenses. Differences in fees are described below, with the Ethiopian currency converted into USD using the 2018 exchange rate to reflect costs following the lifting of the government ban on out-migration in 2018. 

*They spend around one thousand Ethiopian Birr [approximately 36 USD] only. Most of the costs are covered by us [registered recruiters] and it is free for them [migrants]. We receive commission and other expenses from the employer or the agency abroad. They [migrants] cover only the medical, and passport costs*. (Recruiter #1)

*Informal recruiters request about 25,000–30,000 Birr [approximately 900–1000 USD] per individual before the migrants leave the country … that created an enormous burden and challenge to the migrants*.(Stakeholder #9)

Price was, therefore, one of the few differences noted between the use of regular or irregular routes. Fees were not, however, a primary concern when planning migration, as migrants’ extended families usually borrowed money or sold assets such as land or livestock to fund relatives’ departure.

### 3.3. Difficulty Distinguishing Legal versus Irregular Migration 

Accounts from returnees described that there was rarely a clear “choice” between regular versus irregular migration. In most cases, migrants pursued whatever process was presented by their personal contacts, which could include local returnees, licensed recruiters, and unregistered brokers. These different actors in the migration process often operated in similar ways. Because migrants favoured rapid overseas placement, both registered and unregistered agencies sent migrants as quickly as possible, often in ways that were outside the law. Not only do registered and unregistered recruiters appear similar to community members, they sometimes operate in the same way—something acknowledged by many respondents, as described by one returnee: 


*The problem is that agencies that have a legal status are involved in illegal activities. It is difficult for us all to distinguish a legal agency from an illegal one. They all say the same thing.*
(Returnee #1)

Registered agencies are located primarily in larger cities, mostly in the capital city, Addis Ababa. Thus, rural women cannot easily access them without intermediary brokers. As described below, registered agencies use local brokers to identify prospective migrants in rural communities and link them to the agency, although use of such intermediaries is illegal: 


*Usually the migrants do not come to our office directly. There is a third party, a broker, who arranges the deal. The migrants are brought here by the brokers*
(Recruiter #10)

One reason given for why registered employment agencies could flout regulations and operate under the radar was weak government supervision and oversight and a lack of enforcement mechanisms. As noted, over 800 agencies had registered with MOLSA, which seemed unable to monitor each closely. This large number of agencies makes it unlikely that any given agency’s behaviour will be scrutinized, suggesting the system relies on registered recruiters’ professional integrity and willingness to self-govern versus any oversight mechanism. 


*[Registered agencies] make a pledge or a series of formal promises to protect the safety of migrants if they are given a license. Of course, ultimately it is your integrity or the quality of having strong moral principles that determine the outcome, no one will check or control the agencies after licensing*
(Recruiter #10)

Finally, migrants and their families experienced difficulty distinguishing between regular and irregular means of migration because so much of their overall migration experience depended on what happened to them once in their destination country. Returnees often blamed their poor experiences on the power their employers had over them, and the randomness of “fate” that determined the kind of household into which they were placed as domestic workers. 

*Some of my employers have refused to pay me my salary. However, I kept quiet. I could not bring the case to the attention of a court of law/the legal system because I was an illegal migrant. I had neither a passport nor residence permit*.(Returnee #3)

## 4. Discussion 

Despite the introduction of new migration policy measures designed to increase migrants’ safety, many Ethiopians use irregular channels for their labour migration, which has been said to increase an individual’s risk of being trafficked or exploited. This qualitative study examined reasons why women who seek domestic work in the Middle East countries might opt for irregular forms of migration. 

We identified three main themes that help explain the persistence of irregular migration among Ethiopian female migrants. First, irregular migration was seen to have fewer steps and bureaucratic barriers than regular migration procedures, making it both quicker and easier to navigate. Second, while study participants did associate “illegal” migration with greater risks of exploitation, they also recognized that using registered overseas employment agencies did not reliably reduce these risks. They believed the risks might be the same, both because migration outcomes were most dependent on the employer and employment conditions at the destination, over which Ethiopian recruiters had very little control, and because they felt recruiters were always acting in their own financial interests, whether licensed or not. Finally, migrants could not always distinguish between regular and irregular migration as a range of local informal actors were likely to be involved in both types, and both registered and unregistered agents operated in similar ways. 

Our finding that some migrants actively seek out irregular migration because it is seen to be both easier and faster highlights the value put on streamlined procedures by potential migrants. This was mainly because of strict migration requirements introduced by the 2016 Ethiopian Overseas Employment Proclamation (No.923/2016) [24], introduced in response to heightened reports of abuse of Ethiopian domestic workers abroad and mass deportation of Ethiopian migrants from Saudi Arabia [11,25,26]. The 2016 Proclamation tightened licensing requirements for private employment agencies but also added requirements for age, school attainment, and predeparture skills training. Although these requirements were meant to promote safer regular migration, we found they were considered burdensome and time-consuming, influencing migrants to select irregular migration. The 2021 Proclamation removed the eighth grade education attainment requirement in recognition that this forced less educated women, including experienced returnees, to use irregular channels. 

Our findings suggest that other bureaucratic barriers remain, reflecting results of other studies, in which migrants travelling by irregular means face fewer administrative requirements and thus can travel more quickly, or without the requirement to first secure a valid job offer [14,27,28]. Similarly, research in other African countries has also identified the presence of numerous registered (but unregulated) recruitment agencies and brokers despite government efforts to introduce registration to reduce irregular migration [14,25,27,28]. These agents use informal mechanisms, such as relying on social networks to recruit potential workers from their communities, and may have a chain of collaborators all the way up to government officials and diplomats who ease and hasten visa and travel arrangements. This may be partly due to most registered agencies being based in large cities, making them inaccessible to rural residents. A total of 400 listed recruitment agencies that we identified were located in Addis Ababa.

We did not find that Ethiopian migrants select informal recruiters because they are less expensive than formal recruitment agencies, as some previous studies have found [14,27,28]. On the contrary, our respondents told us informal migration facilitators in Ethiopia often charge migrants more money than registered agencies. Under the Ethiopian Overseas Employment Proclamation [22], employers are required to bear the expenses related to their migrant workers’ visa application, round trip transport, residence and work permit application, and insurance. As such, costs incurred by migrants are minimal when their migration is arranged with the help of registered employment agencies, whereas they do not have support from secured employers when using informal facilitators.

One possible incentive for migrants to use registered agents would be if they were believed to offer greater protection and safety than nonregistered agents. However, our findings suggest that although respondents said they believed “legal” migration was safer than “illegal”, there was, nonetheless, prevailing acceptance that there are few real differences in migration outcomes. This result does not support previous research into this area where registered employment agencies were associated with regular and safe migration [9,14,28,29,30,31,32]. We found that registered recruiters also sometimes operate irregularly, negating any difference in migration risk. Although women often consider their migration to have been legal since it occurred with the assistance of registered recruitment agencies and via air travel, in reality, many Ethiopian migrant workers travel using tourist visas and become irregular by overstaying their visas [29]. 

Our most salient findings relate to what happens to migrants at the destination, and how these were not considered related to the type of recruiter in Ethiopia. While recruitment practices are important, until governments or agencies in destination locations can be held accountable for the protection of domestic workers, their safety and well-being will continue to be determined primarily by conditions in the employing household rather than the means by which they got there. This finding reinforced results from previous work conducted among returnees in Ethiopia and elsewhere [8,26,27,33]. 

In Ethiopia, enforcement of ethical recruitment practices is challenged by the high number of registered agencies and the higher number of recruiters that are in demand. It is, therefore, not surprising that migrants have difficulties in distinguishing between regular and irregular migration processes. Prospective migrants’ inability to identify who is providing fully registered and legal services means that promoting legal migration may be a somewhat fruitless exercise. While some migrants knowingly opt for irregular migration, others are uncertain whether or not they are going via the regular or irregular processes. In most cases, it is not so much a specific “choice” between options; rather, migrants just follow the path that is presented through facilitators, to whom they are often referred by individuals they trust, such as returnees, acquaintances, or family members. We also noted during the data collection phase that very few registered agencies had working telephone numbers; thus, migrants must rely on referrals by intermediaries if they want to connect with licensed recruiters. 

Strengths and limitations: A key strength of this study is that we collected data from recruiters, policy and programme stakeholders, and returnees to elicit diverse perspectives. This helped understand why practices by actors across the migration sector can differ from policy intention. These multiple perspectives also helped identify implementation challenges to legislation that has been introduced to help protect labour migrants. 

Our study also has important weaknesses that may affect the interpretation of our findings. The returnees who participated in this study were a specific study population, as they all had been repatriated from Lebanon after being made homeless during the combined crises of Lebanon’s national economic meltdown and the COVID-19 pandemic. These were highly vulnerable migrants requiring emergency assistance and thus likely to differ from other returnees who had worked in other countries and/or arranged their own travel back to Ethiopia. Furthermore, most of these returnees had departed from Ethiopia over three years before the study, meaning they migrated prior to the ratification of new Ethiopian laws and regulations. Their accounts need to be understood within this context. We also found it difficult to locate and interview purposively selected migration recruiters but rather recruited an opportunistic sample of those for whom we were able to find working contact details. Nonetheless, the interviews offer important insights for future recruitment interventions, particularly initiatives that will include especially vulnerable migrants. 

## 5. Conclusions

This study aimed to examine migration decision making by Ethiopian women traveling to the Middle East for domestic labour. Although irregular migration is commonly reported to be a riskier option than going through registered agents, and despite the presence of many registered recruitment agencies, a high number of women use irregular forms of migration. 

The three main reasons for this choice emerging from our study are that (1) irregular migration was perceived to have fewer steps and bureaucratic barriers than regular migration procedures, making it both quicker and easier to navigate; (2) migrants believed risks between irregular and regular routes to be similar; and (3) they could not always distinguish between regular and irregular migration as both registered and unregistered agents operate similarly. 

The development of future initiatives to reduce irregular migration will need to consider that many migrants will opt for irregular mechanisms to avoid perceived bureaucratic barriers that have been introduced to improve safety. As noted, during the timeframe of our study, a new policy amendment was introduced in Ethiopia to reduce unpopular requirements such as eighth-grade education. However, there is an ongoing tension between “aspirational” legal requirements (e.g., women are 18+ years, have completed eighth grade, and undergone predeparture training) and measures that are “realistic”, without delays and administrative barriers. It remains to be seen whether and how new policy measures will reduce the irregular migration of Ethiopian women, which will be an important area for future research. 

## Figures and Tables

**Table 1 ijerph-19-13085-t001:** Characteristics of participants.

Category of Respondents	Total Number of Participants	Composition
Policy and programme stakeholders	18 (12 males and 6 females)	- Eight from federal-level government organisations - Four from regional-level government organizations - Five from local NGOs - One from an international NGO
Recruiters	24 (15 males and 9 females)	- 21 representatives of registered recruitment agencies
Returnee migrants	27 female domestic workers returned from Lebanon	- Age range 21–45 - Number of countries to which they migrated range: 1–3 - Length of time working abroad (in years) range: 1–13

## Data Availability

Due to the sensitive nature of the topic and the disclosure of unlawful practices, we are not making the datasets publicly available. Anyone who would like to access the raw data should contact the researchers directly to discuss potential use.

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
