# Peer review of "Migrant Women’s Health and Safety: Why Do Ethiopian Women Choose Irregular Migration to the Middle East for Domestic Work?"

_ijerph, 2022, doi:10.3390/ijerph192013085_

Round 1

Reviewer 1 Report

The study undertakes to investigate an interesting topic, it examines the background of irregular migration of Ethiopian women. The literature used is relevant, specific and up-to-date on the topic (typically references from the last 5 years).

The strength of the study and also the magnitude of the work done is confirmed by the fact that it explores the opinions of policy stakeholders, migration recruiters, and returnee domestic workers on the topic using a qualitative method (semi-structured interviews). The research work appeared difficult, but it was (strongly) supported by finding and reaching respondents on a sensitive topic.

The importance of the research is confirmed by the item numbers (in total, 66 individual interviews were conducted comprising 18 policy stakeholders, 21 recruiters and 27 returnee migrants from Lebanon).

The authors identified three themes that capture the migration decision-making process (Ease and speed; Perceptions of risk; Difficulty distinguishing regular from irregular migration.)

The presentation of the results is clear and understandable.

Suggestions:

* The returnees who participated in this study were a specific study population, as they all had been repatriated from Lebanon after being made homeless, so I recommend indicating this among the keywords, possibly including it in the title, because in its current form it seems to be general.

* Placing the research limitations paragraph in a separate subsection and supplementing it with possible future directions of the research.

* Development of the conclusion chapter, highlighting the three main decision-making process topics and their results.

* Formulation of more specific action proposals instead of general ones.

* If any specifics/research results are available regarding the effects of the introduction of the new directive amendment, it may be worth mentioning (possible new research direction?)

 Congratulations to the Authors!

Reviewer 2 Report

This very well-organized and well-written paper analyses the reasons Ethiopian women choose irregular vs. regular channels for migration as domestic workers to the Middle East.  Based on 66 semi-structured interviews with representatives of institutions involved in migration and labor issues; legalized migrant labor recruiters; and Ethiopian migrant workers who returned from Lebanon due to Covid disruptions, the authors analyze 3 themes they identified in the decision process:  ease and speed; perceptions of risk; and difficulty distinguishing between regular and irregular migration processes.  The paper identifies bureaucratic barriers to choosing regular migration, and a lack of effective protections after placement by either regular or irregular recruiters.  The findings highlight the ineffectiveness of current Ethiopian policies to regulate migration.

This is a well-focused qualitative study the produces illuminating findings, based on a carefully-described sampling (purposive and snowball) and interview analysis protocol.  I have only one question about the sampling process, which is the specific position of representatives of the “policy stakeholder” organizations sampled.  The authors describe the selection of a range of institutions concerned with migration and labor issues, but do not describe the specific positions of the individuals interviewed within those institutions.  This is important since interviews are carried out with individuals, whose perspectives may vary widely within institutions.  A bit more information about these organizations, and the persons interviewed, would be useful.

One error appears on line 318 when the authors mention “800” organizations listed with MOLSA, while the number is cited as 400 on lines 112 and 375.

Other than these minor errors, this paper is publication-ready, and provides useful insights on the factors affecting the health risks for migration by Ethiopian women to the Middle East.
